# A Novel Time Delay Estimation and Calibration Method of TI-ADC Based on a Coherent Optical Communication System

**Yongjie Zhao, Sida Li, Longqing Li and Zhiping Huang \***

College of Intelligent Science and Technology, National University of Defense Technology, Changsha 410073, China; zhaoyongjie17@nudt.edu.cn (Y.Z.); lisida13@nudt.edu.cn (S.L.); longqing_li@nudt.edu.cn (L.L.)

**\*** Correspondence: huangzhiping65@nudt.edu.cn; Tel.: +86-137-0731-3262

**Abstract:** In optical communication systems, coherent detection is a standard method. The received signal enters the digital domain after passing through a time-interleaved analog-to-digital converter (TI-ADC). However, the time delay of the ADC brings noise into the signal, which decreases the signal quality; therefore, ADC calibration is essential. At present, there are many calibration methods for time delay, but their performances are not satisfactory at a high sampling frequency. This paper presents a method of time delay estimation and calibration in a coherent optical communication system. First, the expected maximum (EM) method is used to roughly estimate the time delay and then transfer the estimated value into the trained back propagation (BP) neural network to generate more accurate results. Second, the sampled signal is reconstructed, and then a finite impulse response (FIR) filter is designed to compensate for the time delay. There are several advantages of the proposed method compared with previous works: the convergence with a BP network is faster, the estimation accuracy is higher, and the calibration does not affect the sample operation of the ADC working in the background mode. In addition, the proposed calibration method does not need additional circuits and its low power consumption provides more sources for dispersion compensation, error correction, and other subsequent operations in the coherent optical communication system. Based on the quadrature phase shift keying (QPSK) system, the proposed method was implemented in a 16-channel/8-bit, 40-GS/s ADC. After estimation and calibration, the relative error of estimation was below 1%, the signal noise distortion rate (SNDR) reached 55.9 dB, the spurious free dynamic range (SFDR) improved to 61.2 dB, and the effective number of bits (ENOB) was 6.7 bits. The results demonstrate that the proposed method has a better calibration performance than other methods.

**Keywords:** optical communication; coherent optical communication system; analog-to-digital converter; back propagation neural network; time delay calibration

## 1. Introduction

In many optical communication infrastructures, in order to meet the broader bandwidth requirements, the sampling frequency and the bandwidth of ADC need to be continuously improved. Researchers have proposed a time-interleaved sampling scheme; the parallel sampling scheme increases the sampling rate to N times that of a single ADC [1–6].

However, multi-channel collaborative sampling introduces mismatches among channels, including offset, gain, time delay, and bandwidth mismatches [7]. The offset and gain mismatches are static errors; channel equalization in the digital domain can quickly weaken these static errors [8,9] Errors caused by a time delay grow when the input frequency is improved, which is dynamic. Compared with static error calibration, time delay calibration is more complicated and researchers have conducted a lot of research on

the subject: The authors of [7,10–12] divided the TI structure into two topological structures, which improves the SNDR in time delay calibration, and the convergence is fast, but the SNDR is unavailable in coherent communication due to the low sampling frequency, and the digital-mixing structure causes crosstalk among channels. The authors of [13] proposed a background calibration method based on a 65 nm, 6-bit, 16-GS/s TI-ADC; the delay loop lock (DLL) is used to generate 8 sampling interfaces as the multi-phase clock-generator and the mismatch is reduced after digital calibration without pre-emphasis. Nevertheless, the structure is complex and the clock jitter can affect the performance of the calibration processor, so it places high demands on the clock. Blind recovery is also used to calibrate the time delay with low complexity [14–17], this method does not need to know the input signal as long as it meets the bandwidth requirement. In addition, the calibration can be operated while the ADC is sampling, but it needs large amounts of oversampling and is not applicable for TI-ADC, which contain more than two channels. For coherent optical communication, the authors of [18] used a low complexity adaptive equalizer to compensate for the mismatches of the TI-ADC; the equalizer was adapted using the stochastic gradient descent (SGD) algorithm. A post-processed version of the error is available at the slicer of the receiver, where the post processing is done using the backpropagation algorithm. However, its relative error is not good, and the convergence is slow. In terms of mismatch estimation, the least squares (LS) method has been widely used [19–22] to fit the distortion components and estimate the mismatch; the uncalibrated codes are corrected with the estimated results and a loop filter. Although the method is efficient and straightforward, there is no iteration to improve the accuracy, the type and bandwidth of signal are limited, and it may be unsuitable in the QPSK coherent optical communication system at a high sampling frequency. The authors of [23] note that adding a single-tone sine wave to the analog inputs calibrates the mismatches, but it is a foreground calibration, and conversion cannot be carried out when calibrating. Besides, the LS method cannot track supply and temperature variations, and the process is complicated with additional analog circuits.

In this paper, a novel calibration method is proposed based on the coherent optical communication system. The block diagram is shown in Figure 1a: The optical signal is converted into the 4-way analog signal after passing through the coherent receiver. The analog signal is transmitted into 4 ADCs to accomplish the analog-to-digital conversion, and the background calibration starts after the ADCs launch. Figure 1b shows the flow chart of the ADC calibration without any additional analog circuits. First, the EM method is applied to roughly estimate the time delay $\Delta t$ and a BP neural network is built to make further estimates; after being processed by the network, the estimation accuracy improves and outputs a more accurate time delay $\Delta t'$. Second, the signal is reconstructed based on $\Delta t'$ and an FIR filter is designed to compensate for the reconstructed signal. Finally the filter outputs the calibrated signal. Experiments were carried out in the 112 Gbps QPSK coherent optical communication system with a 45 nm 40-GS/s TI-ADC, whose bandwidth was 16 GHz. The relative error of time delay estimation was below 1% with fast convergence, which is smaller than that of the LS method; after calibration, the SNDR reached 55.9 dB, the SFDR improved to 61.2 dB, and the ENOB reached 6.7 bits, which are better results than those obtained in previous works.

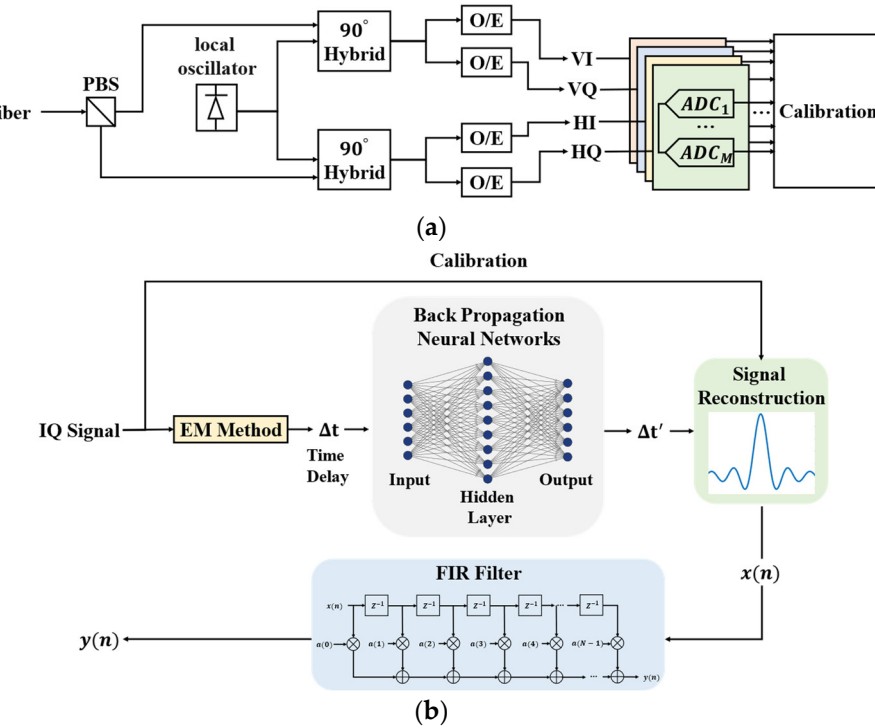

**Figure 1.** (**a**) The diagram of the optical coherent communication system and the role the ADC played in the system; (**b**) the flowchart of the calibration method.

Section 2 reviews the LS estimation method. Section 3 describes the BP neural network combined with the EM method and calibration based on signal reconstruction with an FIR filter. Section 4 verifies the reliability and advantages of the proposed method through experiments and makes a comparison with the state-of-the-art TI-ADCs. Finally, Section 5 provides some conclusions.

## 2. Review of the Least Square Estimation Method

After being processed by the coherent receiver, the input of the ADC can be represented as:

$$x(t) = \alpha \cos(2\pi f_{in} t + \phi) + C \tag{1}$$

where $\alpha$, $f_{in}$, $\phi$, $C$ are the amplitude, frequency, phase, and direct current (DC) offset of the input signal, respectively. The ADC outputs the sampling sequence $y[n]$ with the sampling frequency at $f_s$. After the least square fitting, the sequence can be represented as $\hat{y}[n]$:

$$\hat{y}[n] = \hat{\alpha} \cos(\omega n + \hat{\phi}) + \hat{C} = \hat{A} \cos(\omega n) + \hat{B} \sin(\omega n) + \hat{C}, n = 0,1,2,\dots N-1 \tag{2}$$

where $\hat{\alpha}, \hat{\phi}, \hat{C}$ are the estimated amplitude, phase, and DC offset of the signal, respectively, $\hat{A} = \hat{\alpha} \cos(\hat{\phi}), \hat{B} = -\hat{\alpha} \sin(\hat{\phi}), \omega = 2\pi f_{in}/f_s$, and $N$ is the length of the sampling sequence. In the ideal case, $ADC_i$ can be understood as $ADC_0$ after interpolation without time delay among $M$ channels. This article uses a Chebyshev I-type interpolation filter to generate the ideal sequence of channel $i$ $(i = 0,1,\dots,M-1)$:

$$\bar{y}_i[n] = \bar{A}_i \cos(\omega n) + \bar{B}_i \sin(\omega n) + \bar{C}_i, n = i, i+M, \dots, i+N-M \tag{3}$$

where $\bar{A}_i = \bar{\alpha}_i \cos(\bar{\phi}_i), \bar{B}_i = -\bar{\alpha}_i \sin(\bar{\phi}_i)$. Due to the time delay in the channel, the actual output sequence can be represented as:

$$y_i[n] = P_i \cos(\omega n) - Q_i \sin(\omega n) + \bar{C}_i, n = i, i+M, \dots, i+N-M \tag{4}$$

$$P_i = [\bar{A}_i \cos(2\pi f_{in}\Delta t_i) + \bar{B}_i \sin(2\pi f_{in}\Delta t_i)]$$

$$Q_i = [\bar{A}_i \sin(2\pi f_{in}\Delta t_i) - \bar{B}_i \cos(2\pi f_{in}\Delta t_i)]$$

where $\Delta t_i$ is the time delay of channel $i$. (2) and (4) both represent the actual output of ADC; $\Delta t_i$ is obtained according to this equivalence:

$$\Delta t_i = \frac{\tan\left(\frac{\hat{A}_i\bar{B}_i - \bar{A}_i\hat{B}_i}{\hat{A}_i\bar{A}_i + \bar{B}_i\hat{B}_i}\right)}{2\pi f_{in}} \tag{5}$$

### 3. Estimation with the EM-BP Method and Signal Reconstruction with FIR

#### *3.1. Time Delay Estimation*

The LS estimation method is widely used for signal processing, but it has no iteration to approach the actual time delay; consequently, the estimation accuracy is not satisfied at a high sampling frequency, up to 40-GS/s. The proposed method combines the EM with a BP neural network to compensate for this deficiency; the EM method is utilized first and then the results are transferred to the BP neural network for further estimation. Compared with the LS method, the proposed method has a feedback mechanism to increase the accuracy; besides, the method can estimate a broader range of time delays than can the LS method, which is more adaptable in practical use.

#### 3.1.1. Expectation Maximum Method

The output signal can be regarded as a multi-tone signal composed of $Mq$ signals, which can be represented as ($q$ is the number of frequency components):

$$s_m = A_m \cos(\omega_m n) + B_m \sin(\omega_m n) + C_m$$
$$m = 0,1,2,\ldots,Mq - 1; n = 0,1,2,\ldots,N - 1 \tag{6}$$

where $A_m = \alpha_m \cos(\phi_m), B_m = -\alpha_m \sin(\phi_m),$ $\alpha_m,$ $\phi_m$ and $C_m$ are the amplitude, phase, and DC offset, respectively, $\omega_m = 2\pi f_m/f_s,$ $f_m$ is the frequency of $m^{th}$ signal. The actual sampling sequence can be represented as:

$$y[n] = \sum_{m=0}^{Mq-1} y_m[n] = \sum_{m=0}^{Mq-1} (s_m[n; \theta_m] + e[n]) \tag{7}$$

where $\theta_m = [A_m, B_m, C_m],$ $e[n]$ is the quantization noise, and the maximum likelihood function $L(\theta)$ can be represented as:

$$L(\theta) = c - \frac{1}{2}\sum_{n=0}^{N-1}\left(y[n] - \sum_{m=0}^{Mq-1} y_m[n; \theta_m]\right)^* \cdot \Phi^{-1} \cdot \left(y[n] - \sum_{m=0}^{Mq-1} y_m[n; \theta_m]\right) \tag{8}$$

where $\Phi = E(e[n] \cdot e[n]^T),$ $E$ represents the expectation, and $c$ is a constant. The specific steps of the algorithm are shown in Appendix A; the algorithm converges, $L(\theta)$ reaches a maximum, and $\Delta t_i$ can be calculated.

#### 3.1.2. Back Propagation Neural Network

In order to increase the estimation accuracy, this article built a BP neural network, which was formed by connecting multiple neurons according to specific rules, as shown in Figure 2. The network included an input layer, a hidden layer (containing 100 neurons), and an output layer. The "+1" represents the bias, which improves the fitting accuracy. The samples sent to the network included the EM estimation results and the true values of the time delay. The proportions of the training set, validation set, and test set were 70%, 15%, and 15%, respectively. The number of epochs was 100, and there were 50 iterations per epoch. The samples propagated forward in the supervised learning process with a

learning rate at 0.01. There was an error between the output and the expected result after each training when the error was greater than 0.01 ps, and the error was propagated back to change the weight $w$ between the layers. We kept training the network until the error was below 0.01 ps or the network accomplished 100 epochs.

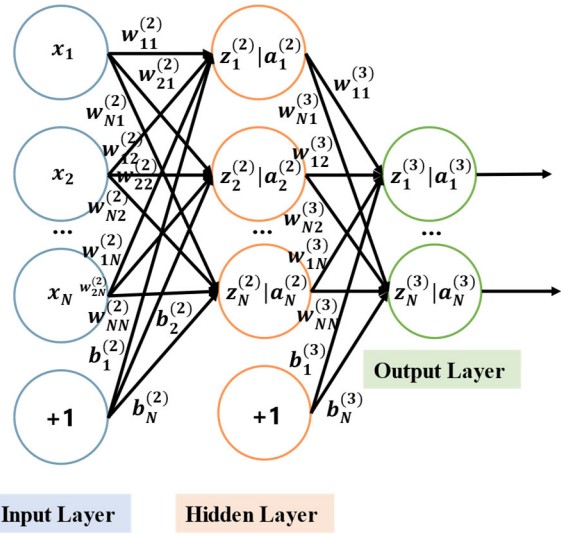

**Figure 2.** A typical BP neural network structure includes an input layer, a middle layer (hidden layer), and an output layer.

The implementation process of the network can be divided into the following steps:

Step 1. Normalize the true values and the estimated results of the time delay. Assign a random number in the interval $(-1, 1)$ to each connection weight $w$, set the error function $e$, provide the calculation accuracy value $\varepsilon$, and the maximum number of learning $M$.

Step 2. Select the $k^{th}$ estimated result and the corresponding expected output randomly:

$$x(k) = \left( x_1(k), x_2(k), \ldots, x_n(k) \right)$$
$$d_0(k) = \left( d_1(k), d_2(k), \ldots, d_n(k) \right) \tag{9}$$
$$k = 1, 2, \ldots, m$$

where $x(k)$ is the input training set, $m$ is the number of samples, $n$ is the number of neurons of the input layer, and $d(k)$ is the expected output.

Step 3. Calculate the input and output of each neuron in the hidden layer:

$$hi_h(k) = \sum_{i=1}^{n} w_{ih} x_i(k) - b_h, h = 1, 2, \ldots, p$$
$$ho_h(k) = f\left( hi_h(k) \right), h = 1, 2, \ldots, p$$
$$yi_o(k) = \sum_{h=1}^{p} w_{ho} ho_h(k) - b_o, o = 1, 2, \ldots, q \tag{10}$$
$$yo_o(k) = f\left( yi_o(k) \right), o = 1, 2, \ldots, q$$

where $hi$ and $ho$ are the input and output of the hidden layers, respectively, $yi$ and $yo$ are the input and output of output layers, respectively, $w_{ih}$ is the connection weight of the input layer and the middle layer, $w_{ho}$ is the connection weight of the hidden layer and the output layer, $b_h$ is the threshold of each neuron in the hidden layer, $b_o$ is the

threshold of each neuron in the output layer, $k$ is the number of samples, $f(\cdot)$ is the activation function, $p$ and $q$ are the number of neurons of the hidden layer and output layer, respectively, and error function $e$ can be represented as:

$$e = \frac{1}{2}\sum_{o=1}^{q}(d_o(k) - yo_o(k))^2 \tag{11}$$

Step 4. Use the expected output and actual output of the network to calculate the partial derivative of the error function $\delta_o$ to each neuron in the output layer and $\delta_h$ to each neuron in the hidden layer:

$$\delta_o(k) = (d_o(k) - yo_o(k)) \cdot f'(yi_o(k)) \tag{12}$$

$$\delta_h(k) = \left(\sum_{o=1}^{q}\delta_o(k)w_{ho}\right)f'(hi_h(k)) \tag{13}$$

Step 5. Modify the weights between the connected layers:

$$w_{ho}^{N+1} = w_{ho}^{N} + \mu\delta_o(k)ho_h(k)$$
$$w_{ih}^{N+1} = w_{ih}^{N} + \mu\delta_h(k)x_i(k) \tag{14}$$

where $\mu$ is the learning rate of the network.

Step 6. Calculate the global error:

$$E = \frac{1}{2m}\sum_{k=1}^{m}\sum_{o=1}^{q}(d_o(k) - y_o(k))^2 \tag{15}$$

Step 7. Determine whether the network error meets the requirements. When the error is below 0.01 ps or the network reaches 100 epochs, the learning process is complete; otherwise, select the next batch and the corresponding expected output, return to Step 3, and enter the next round of learning.

The time and space complexity of the neural network can be represented as:

$$Time \sim O_t(M^2 \cdot K^2 \cdot \sum_{l=1}^{D}C_{l-1} \cdot C_l)$$

$$Space \sim O_s(\sum_{l=1}^{D}K^2 \cdot C_{l-1} \cdot C_l + \sum_{l=1}^{D}M^2 \cdot C_l) \tag{16}$$

where $M$ is the side length of the output feature map, $K$ is the size of input samples, $C$ is the number of channels, and $D$ is the number of layers. In this network, $M = 1$, $K = 1 \times 1000$, $D = 2$, $C_0 = C_2 = 1$, and $C_1 = 100$, thus the time complexity is $O_t(2 \times 10^8)$, the space complexity is $O_s(2 \times 10^8 + 101)$, and according to the complexity theory, the space complexity can be approximated as: $O_s(2 \times 10^8)$.

*3.2. Calibration Based on Signal Reconstruction with FIR*

In previous works, a known signal was added to obtain the unknown time delay, which is a complex process requiring additional analog circuits; subsequent mixture with the target signal reduces the resolution. Calibration can also be conducted based on blind recovery, although the complexity of this method is low, it needs large amounts of oversampling and is not applicable for TI-ADC, which contains more than two channels. Considering the shortcomings of the above methods, an FIR was designed to compensate for the reconstructed signal without any analog circuits, and it produces no restrictions to the number of channels.

Periodic non-uniform sampling theory points out that as long as the average sampling rate of the system is higher than the Nyquist rate, as in the case of an ideal low-pass

filter, the sampling signal can be reconstructed [24]; however, the ideal low-pass filter does not exist in practice. To reconstruct the sampled signal, the sampling frequency should usually be at least 2.56 times greater than the highest frequency component of the input signal. In the 112 Gbps QPSK coherent optical communication system, a sampling frequency at 40-GS/s can meet that requirement. The detailed derivation of the method is as follows. The actual output with time delay can be represented as:

$$y[n] = x(nT_s + \Delta t_i) = \sum_{m=0}^{N-1} x[m] sinc((n-m) + \frac{\Delta t_i}{T_s}), i =< n, M > \tag{17}$$

where $x(t)$ is the input signal, $x[m]$ is the reconstructed signal, $sinc$ is the sampling function, $M$ is the number of channels, $N$ is the length of $y[n]$, $\Delta t_i$ is the time delay estimated by the proposed method described in Section 3.1, $T_s = 1/f_s$, expand $sinc$ into a higher-order Taylor expansion, $y[n]$ can be represented in a convolutional form:

$$y[n] = x[n] + \sum_j \frac{1}{j!} \left( \frac{\Delta t_i}{T_s} \right)^j \left( h_j * x \right)[n] \tag{18}$$

where $h_j$ is the $j^{th}$ order partial derivative of $sinc$, which can be represented as:

$$h_j[n] = \frac{\partial^j}{\partial t^j} sinc(t/T_s), t = nT_s \tag{19}$$

The second-order expansion is carried out for $y[n]$, and the Fourier transform of $y[n]$ can be represented as:

$$Y(e^{j\omega}) \approx X(e^{j\omega}) + \frac{\Delta t_i}{T_s} H_1(e^{j\omega}) X(e^{j\omega}) + \frac{1}{2} \left( \frac{\Delta t_i}{T_s} \right)^2 H_2(e^{j\omega}) X(e^{j\omega}) \tag{20}$$

where $X(e^{j\omega})$ is the Fourier transform of $x(t)$, $H_1(e^{j\omega})$ and $H_2(e^{j\omega})$ are the Fourier transforms of $h_1$ and $h_2$ respectively, so $X(e^{j\omega})$ can be represented as:

$$X(e^{j\omega}) \approx \frac{Y(e^{j\omega})}{1 + \frac{\Delta t_i}{T_s} H_1(e^{j\omega}) X(e^{j\omega}) + \frac{1}{2} \left( \frac{\Delta t_i}{T_s} \right)^2 H_2(e^{j\omega})} \tag{21}$$

After the second-order expansion for (21), $X(e^{j\omega})$ can be represented as (for ease of presentation, $e^{j\omega}$ is omitted):

$$X \approx Y - YH_1 - \frac{1}{2} \left( \frac{\Delta t_i}{T_s} \right)^2 (YH_2 - 2YH_1^2) \tag{22}$$

After Fourier inverse transform, the reconstructed signal can be represented as:

$$x[n] \approx y[n] - \frac{\Delta t_i}{T_s} (y * h_1)[n] - \frac{1}{2} \left( \frac{\Delta t_i}{T_s} \right)^2 ((y * h_2)[n] - 2(y * h_1 * h_1)[n]) \tag{23}$$

With the reconstructed signal, an FIR filter with 20 taps is employed to further compensate for the time delay, the FIR provides feedback multiple times to approach the expected signal, the relationship between the FIR and the input sequence $x[n]$ are given in the form of a finite convolution sum, and each tap needs a multiplier accumulator (MAC) unit that consumes logic resources. The input signal is temporal and changes with time. When the maximum eigenvalue is $\lambda_{max}$ and convergence factor $\mu = 1/\lambda_{max}$, the input of $M$ taps can be represented as:

$$x_{in} = [x_k, x_{k-1}, \dots, x_{k-M+1}], k = M - 1, \dots, N - 1 \tag{24}$$

The filter output can be represented as:

$$z_k = W_{k-1} \times x_{in} \tag{25}$$

where $W_{k-1}$ is the weight. The error of FIR is:

$$e_k = d_k - z_k \tag{26}$$

where $d_k$ is the expected output. As a result, the weight can be represented as:

$$W_k = W_{k-1} + 2\mu e_k x_{in} \tag{27}$$

By bringing (27) into (25) to update $z_k$, the FIR converges and outputs $z_k$ after $e_k$ is lower than the maximum allowable error.

## 4. Experimental Results

The experiment was based on the 112 Gbps QPSK coherent optical communication system, and the fiber length was 100 km; the power and wavelength of the continuous wave (CW) laser were 2 dBm and 1550 nm, respectively. The analog-to-digital conversion was implemented in a 45 nm CMOS, 16-channel/8-bit, 40-GS/s ADC, with the bandwidth at 16 GHz.

### 4.1. Comparison of Estimation Accuracy among LS, EM, and EM-BP Neural Network

The estimation accuracy affects the calibration, so a simulation was conducted to verify the validity of the estimation algorithm before the experiment. The conditions of the simulation were the same as above. The relative error $\delta_i$ of estimation can be represented as:

$$\delta_i = \frac{\Delta t_i - T_i}{T_i} \times 100\% \tag{28}$$

where $\Delta t_i$ is the estimated time delay of channel $i$ and $T_i$ is the true value of time delay, which is added manually before the simulation. The results are shown in Figure 3; estimations can be completed at input frequencies up to 16 GHz, which is available in the actual 112 Gbps QPSK system. The LS and EM method results were below 100%, and the relative error of estimation based on the EM-BP neural network was below 1% in all channels. Figure 3d shows the average relative error of 16 channels with the three methods, and the EM-BP method was more accurate than the other two methods. In addition, the speed of convergence was faster and the number of iterations was just 5000, which is less than the authors of [18] found with $10^5$ iterations. As a result, the simulation validated the proposed method, and the high estimation accuracy provided a favorable condition for the subsequent calibration.

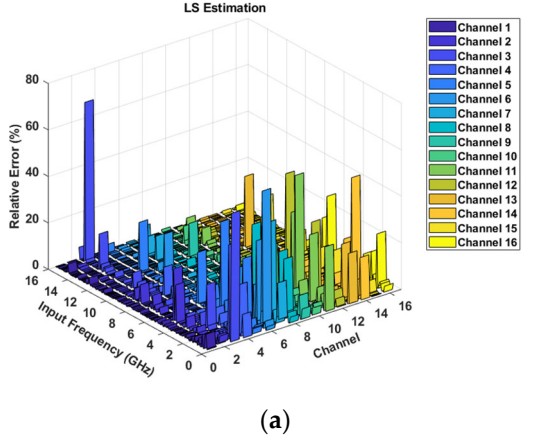

(**a**)

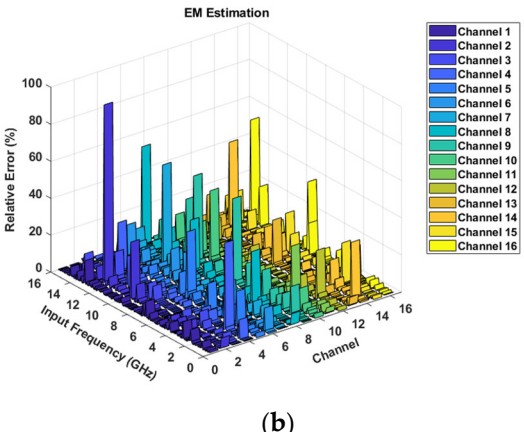

(**b**)

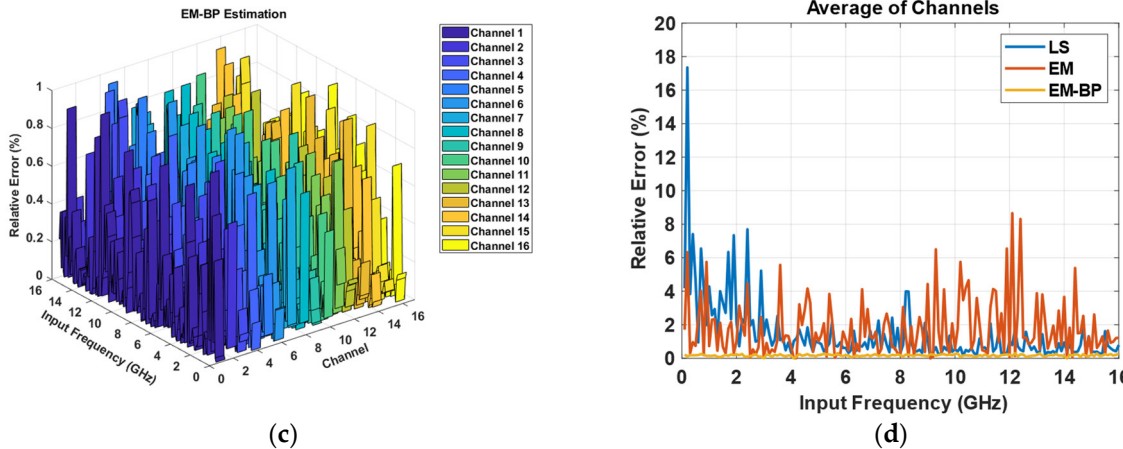

(**c**)　　　　　　　　　　　　　　　　　　　　(**d**)

**Figure 3.** Relative error comparison among LS, EM, and EM-BP methods; the sampling frequency was 40-GS/s. (**a**) Relative error of LS estimation; (**b**) relative error of EM estimation; (**c**) relative error of estimation based on the EM-BP method; (**d**) average results of 16 channels; the relative error of the EM-BP method was the lowest and was below 1%.

### 4.2. Calibration of the Time Delay

Calibration started after the estimation of time delay; the curves of SNDR with a time delay $\Delta t$ are shown in Figure 4. The signal was reconstructed first: The SNDR improved when $\Delta t < 12$ ps, but when $\Delta t > 12$ ps, the SNDR decreased instead. The reconstructed signal was transmitted to the FIR, and the SNDR improved again. With lower values of $\Delta t$, the reconstruction performance was better, and the calibrated signal was closer to the ideal signal. Although the curve decreased as $\Delta t$ grew, there was still an improvement compared with the original signal.

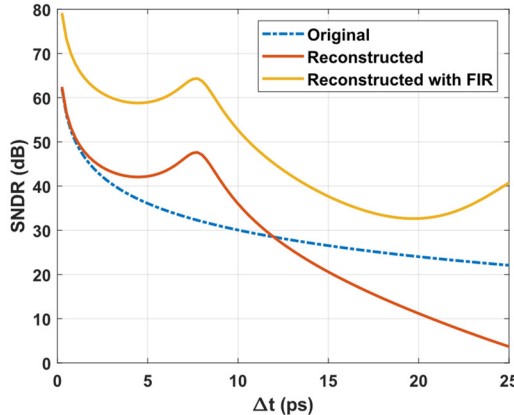

**Figure 4.** SNDR curves in different situations with changes in the time delay. First the signal was reconstructed and then it was transmitted to the FIR; this process further improved the SNDR.

Figure 5 shows the output spectrum of the signal before and after the calibration, the tones of time delay and other harmonics were weakened after calibration and the SNDR improved from 29.0 dB to 55.9 dB. The signal quality improved, which laid a foundation for dispersion compensation, carrier recovery, and decision operations in the QPSK system.

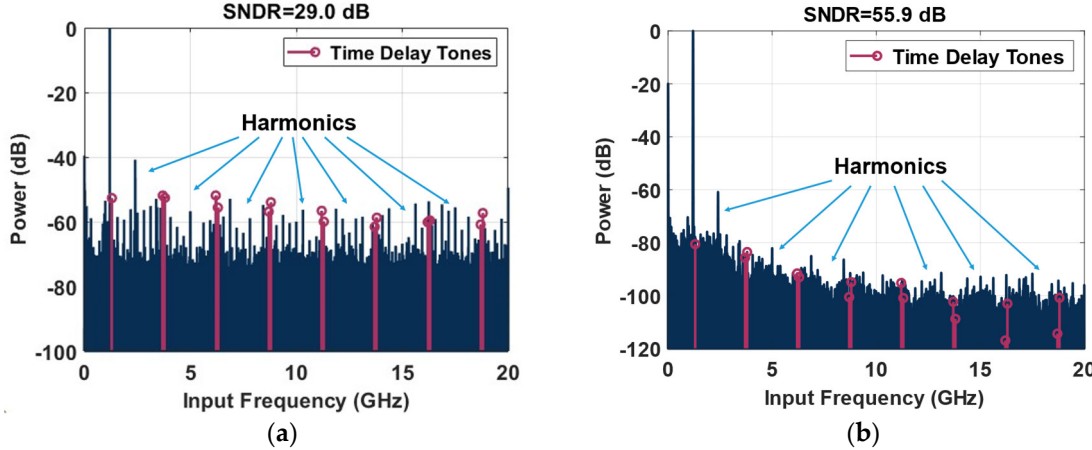

**Figure 5.** (**a**) The spectrum before calibration; (**b**) the spectrum after calibration.

This work was compared with some state-of-the-art TI-ADCs, and a summary is shown in Table 1. This work has advantages in terms of channels and sampling frequency; the SNDR, SFDR, and ENOB are the highest at sampling frequencies above 5-GS/s. The performance of the method from [11] is the best of the methods listed in Table 1; however, the TI-ADC is not suitable in the coherent optical transmission system with the sampling frequency at 1.6-GS/s. The formulas of SNDR, SFDR, and ENOB are shown in Appendix B.

**Table 1.** Comparison of the proposed method with some state-of-the-art TI-ADCs.

| Publication | This Work | [13] | [23] | [10] | [11] |
|---|---|---|---|---|---|
| Technology (nm) | 45 | 65 | 65 | 28 | 28 |
| Resolution (bit) | 8 | 6 | 6 | 10 | 10 |
| TI Channels | 16 | 8 | 16 | 16 | 8 |
| Sampling Frequency (GS/s) | 40 | 16 | 40 | 5 | 1.6 |
| Supply (V) | 1.0 | 1.5 | 1.0/2.5 | 1.0/0.85 | 0.9/0/8 |
| SNDR (dB) | 55.9 | 28.0 | N.A. | 48.5 | 54.2 |
| SFDR (dB) | 61.2 | 40.4 | 35.0 | 59.6 | 67.1 |
| ENOB (bits) | 6.7 | 4.9 | 5.5 | N.A. | N.A. |

## 5. Conclusions

This article presents a method to calibrate the time delay of a TI-ADC in a coherent optical communication system. The method includes two steps: estimate the time delay and then reconstruct the signal with the FIR filter based on the estimated results. The EM-BP method showed higher accuracy (relative error was below 1%) and faster convergence compared with those of previous works. Without additional analog circuits and limitation of the input signal, the 16-channel 40-GS/s 8-bit TI-ADC reached an SFDR of 61.2 dB, an SNDR of 55.9 dB, and increased the ENOB from 4.1 bits to 6.7 bits. These important results demonstrate that this work has better performance and a competitive position in the state-of-the-art TI-ADCs. Furthermore, ADC calibration plays an essential role in the QPSK system, laying a foundation for subsequent operations.

**Author Contributions:** Conceptualization, Y.Z.; methodology, S.L.; software, Y.Z.; supervision, Z.H.; validation, S.L.; writing (original draft), Y.Z.; writing, review and editing, S.L, investigation, L.L., formal analysis, L.L. All authors have read and agreed to the published version of the manuscript.

**Funding:** This research was funded by Natural Science Foundation of China, grant number 51575517.

**Data Availability Statement:** The data that support the findings of this study are available on request from the corresponding author. The data are not publicly available due to privacy or ethical restrictions.

**Acknowledgments:** The authors would like to acknowledge the Acela Micro Co., Ltd. for providing funding and technical support for the experiments.

**Conflicts of Interest:** The authors declare no conflict of interest.

**Appendix A**

Step 1. Calculate the initial value of $\theta_m$:

$$\theta_m = \begin{bmatrix} A_m \\ B_m \\ C_m \end{bmatrix} = \frac{1}{N} \begin{bmatrix} 2 \cdot Re\{Y_m\} \\ -2 \cdot Im\{Y_m\} \\ \dfrac{1}{Mq} Y_0 \end{bmatrix}, m = 0, \ldots, Mq - 1 \tag{A1}$$

where $Y_m$ represents the FFT calculation of $y_m$, $Y_0$ is the DC offset of $Y_m$, $Re\{\cdot\}$ and $Im\{\cdot\}$ represent the real and the imaginary parts, respectively.

Step 2. Estimate $\hat{y}_m$ through $\theta_m$:

$$\hat{y}_m = D_m \theta_m + \beta_m [y_m - \sum_{m=0}^{Mq-1} D_m \theta_m] \tag{A2}$$

where $D_m$ and $\beta_m$ can be represented as:

$$D_m = \begin{bmatrix} \cos(\omega_m \cdot 0) & \sin(\omega_m \cdot 0) & 1 \\ \cos(\omega_m \cdot 2) & \sin(\omega_m \cdot 2) & 1 \\ \cos(\omega_m \cdot 2(N-1)) & \sin(\omega_m \cdot 2(N-1)) & 1 \end{bmatrix} \tag{A3}$$

$$\beta_m = \frac{\sqrt{A_m^2 + B_m^2 + C_m^2}}{\sum_{m=0}^{Mq-1} \sqrt{A_m^2 + B_m^2 + C_m^2}} = \frac{\sqrt{\theta_m^T \theta_m}}{\sum_{m=0}^{Mq-1} \sqrt{\theta_m^T \theta_m}} \tag{A4}$$

Step 3. Calculate $\theta'_m$ by $\hat{y}_m$:

$$\theta'_m = (D_m^T D_m)^{-1} D_m^T \hat{y}_m \tag{A5}$$

Update $\hat{y}_m$ according to (A2), repeat Step 2–3 until convergence, then start Step 4.

Step 4. Calculate $\Delta t_i$ by $\omega_0, \theta_m$ and $\alpha[k]$:

$$\alpha[k] = \sum_{m=kq}^{(k+1)q-1} \alpha_m \tag{A6}$$

$$\alpha_m = \sqrt{A_m^2 + B_m^2} e^{-j\phi_m}, \phi_m = -\arctan(B_m/A_m)$$

$\alpha[k]$ can also be represented as:

$$\alpha[k] = \frac{1}{j2M} \sum_{i=0}^{M-1} e^{-j\omega_0 \Delta t_i / T_s} e^{-\frac{2\pi jki}{M}} \tag{A7}$$

Combine Equation (A6) with Equation (A7) to determine $\Delta t_i$. We only need to determine the corresponding $\theta'_m$ when $L(\theta)$ reaches the maximum to find $\Delta t_i$, so the value of the constant $c$ can be unclear.

**Appendix B**

$$SFDR = 20 \log_{10} \left( \frac{: X_{avm}(f_i)|}{\max\limits_{f_s, f_h} \{|X_{avm}(f_h)| \ or \ |X_{avm}(f_s)|\}} \right) \quad (A8)$$

where $X_{avm}$ is the averaged spectrum of the ADC output, $f_i$ is the input signal frequency, $f_h$ and $f_s$ are the frequency of the set of harmonic and spurious spectral components, respectively.

$$SNDR = \frac{A_p \sqrt{M(M-3)}}{\sqrt{2 \sum_{m \in S_0} X_{avm} |f_m|^2}} \quad (A9)$$

where $A_p$ is the peak amplitude of the input signal, $M$ is the number of samples in the record, $S_0$ is the set of all integers between 1 and $M-1$, excluding the two values that correspond to the fundamental frequency and the zero-frequency term, and $f_m$ is the $m^{th}$ spectral component.

$$ENOB = \frac{FSR \cdot \sqrt{M(M-3)}}{G \cdot \sqrt{12 \cdot \sum_{m \in S_0} X_{avm} |f_m|^2}} \approx N - \log_2 \left( \frac{\sqrt{\sum_{m \in S_0} X_{avm} |f_m|^2}}{\varepsilon_Q \sqrt{M(M-3)}} \right) \quad (A10)$$

where $FSR$ is the specified full-scale range of the ADC, $G$ is the measured gain (nominally = 1), $N$ is the specified number of bits in the ADC, and $\varepsilon_Q$ is the rms ideal quantization error.

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
