# Peer review of "A Novel Time Delay Estimation and Calibration Method of TI-ADC Based on a Coherent Optical Communication System"

_photonics, doi:10.3390/photonics8090398_

Round 1
Reviewer 1 Report
The paper is not well written and this prevents this reviewer from recommending its publication under this form. Lots of writing mistakes and typos make the paper not attractive. Many variables in the equations are not explained and this makes the readers hard to understand the principle and the novelty of the proposed idea.
My comments are as follows:
1, a flow chart is required to describe the proposed calibration method.
2, P2, Eq.1 to Eq.3, what’s the meaning of ,φ,gi and fin?
3, Line 78, the expression of A and B are incorrect.
4, Lin 95, where is the expression of Ps(s;θ)? Is it a pdf or a likelihood function?
5, Line 147, finite impulse response of FIRs can also have feedback.
6, There is no describe about the coherent optical systems in the experiment.
7, How many nerve cells are used in the BP neural network? and how about the complexity of the neural network?
8, what is the tap number of the FIR?
9, How is the SNR tested or calculated? What is the different between SNDR AND SNR?
10, How is the value of SFDR and ENOB in table 1 obtained? the two variables are not mentioned in the other parts of the manuscript.
Author Response
Dear Reviewer:
We feel appreciated for your patience and constructive comments on our manuscript entitled “A Novel Time Delay Estimation and Calibration Method of TI-ADC Based on Optical Coherent Transmission System” (ID: photonics-1348723). As you pointed, there are still several errors and doubts need to be clarified. According to your nice suggestions, we have made extensive modifications on our previous draft, the detailed corrections in the manuscript are highlighted in blue, and the replies are listed in the attachment which are highlighted in red. Please see the attachment.
Once again, we feel great thanks for your professional review work on our manuscript which would help us to improve the quality of the manuscript both in English and in depth. If there are any other modifications we could make, we would like very much to modify them as soon as possible.
Best regards,
Yongjie Zhao,
Sida Li,
Longqing Li,
Zhiping Huang

Reviewer 2 Report
The paper "A Novel Time Delay Estimation and Calibration Method for TI-ADC Based on Optical Coherent Transmission System" proposes to use a neural network to improve the time delay estimation at the Rx after passing through an ADC.
The paper is generally well structured and scientifically sound. However, the authors should improve on a number of aspects before final acceptance of the paper.
- it is unclear to the reviewer, why in Fig. 1 a "+1" is required as last input in the input and hidden layers. Could you clarify this?
- In Fig. 2 the relative error is plotted over the channel number. The y-axis should be labeled "relative error" to be consistant with Fig. 3. Furthermore, it would be nice to also plot a relative error over input frequency to be able to compare to Fig. 3
- In Fig. 3 the color coding (green, blue, yellow) is unclear to the reviewer. Could you add an appropriate legend? Why did you only plot frequencies up to 10 GHz and not above (which should be possible with a sampling rate of 40 GSa/s). What is the bandwidth of the ADC? Potentially it would be better to show an additional plot of one selected channel over input frequency (see also my comment for Fig. 2) as this would be easier for the reader to exactly quantify the values.
- What is the exact configuration of the neural network used in this study?
Author Response

(The authors gave the same response as above.)

Reviewer 3 Report
Dear Authors,
while I respect the hard work behind your manuscript, in the present form it is very unclear. The correct context for your research is not provided, therefore the aim of your work is not effectively described and even if a contribution to the state of art is mentioned, it is not defended properly. There is a number of language issues, in some cases acronyms are not defined and there are errors in the usage of basic mathematical functions. Unfortunately, the abovementioned issues bring your manuscript below the minimum quality required for an international journal, such as Photonics.
I provide here some feedback about some of the major issues I identified for specific parts of the manuscript.
The introduction does not clearly explain the problem of time delay calibration nor optical coherent transmission systems operation, perhaps a block diagram would have helped to understand the role played by the ADC. The discussions in lines 64-69 should have appeared in the conclusions.
Section 2 is very unclear, and at line 78 there is a mistake in the usage of the formulas for the cosine of a sum.
Section 3 dives in the details of the mathematics, but it fails in the delivering the message. What is really the difference between the solution you proposed with respect to the state of the art? Why your approach is expected to perform better? At line 140, the statement about the Nyquist rate is incomplete, as that would be true only in case of an ideal low-pass filter.
In section 4 and figure 2b, there are some “relative” errors which are above 1. If they can exceed 1, they cannot be relative, therefore a scale for comparison should be given. At line 179, you state that figures 4 shows the impact on filtering on SNR and this in turn proves “the novelty of this article”. I understand the good intention behind with this statement, but I disagree with it and I believe it might be off-putting to some readers. Table 1 presents a comparison of the ADC used for this work with “state-of-art solutions”, but it is not commented in the text. The conclusions are simply a repetition of the introduction and some figures from Table 1, while they should highlight the most important result of your work.
Author Response

(The authors gave the same response as above.)

Round 2
Reviewer 2 Report
The authors have taken up all my suggestions. I consider the manuscript ready for publication.
Reviewer 3 Report
Dear Authors,
thank you for the very good quality of the
modifications, the manuscript has been
significantly improved.